# Space Charge Characteristics and Breakdown Properties of Nanostructured SiO_2_/PP Composites

**DOI:** 10.3390/polym15132826

**Published:** 2023-06-26

**Authors:** Guang-Wei Zhang, Jun-Guo Gao, Ran Wang, Ting-tai Lee, Uwe Schachtely, Hitoshi Kobayashi, Wei-Wang Wang

**Affiliations:** 1Key Laboratory of Engineering Dielectrics and Its Application, Ministry of Education, School of Electrical and Electronic Engineering, Harbin University of Science and Technology, Harbin 150080, China; zhangguangwei9805@163.com (G.-W.Z.); wangran12122022@163.com (R.W.); 2Evonik Specialty Chemicals Co., Ltd., Shanghai 201108, China; tingtai.lee@evonik.com; 3Evonik Operations GmbH, 63457 Hanau-Wolfgang, Germany; uwe.schachtely@evonik.com; 4Evonik (SEA) Pte Ltd., Singapore 138567, Singapore; hitoshi.kobayashi@evonik.com; 5State Key Laboratory of Electrical Insulation and Power Equipment, Xi’an Jiaotong University, Xi’an 710049, China; weiwwang@xjtu.edu.cn

**Keywords:** polypropylene, nanocomposite, surface hydrophobic treatment, space charge, DC breakdown strength

## Abstract

Polypropylene (PP) has gained attention in the industry as an environmentally friendly material. However, its electrical properties are compromised due to space charge accumulation during operation, limiting its application in high-voltage DC cable insulation. This study investigates the effect and mechanism of SiO_2_ with a DDS surface hydrophobic treatment on space charge suppression and the electrical properties of PP composites. The PP matrix was doped with SiO_2_ nanostructures, both with a DDS surface hydrophobic treatment and untreated as a control group. The functional group structure and dispersion of nanostructured SiO_2_ in the matrix were characterized. The findings reveal that the incorporation of SiO_2_ nanostructures effectively mitigates charge accumulation in PP composites. However, a high concentration of unsurfaced nanostructures tends to agglomerate, resulting in inadequate space charge suppression and a diminished DC breakdown field strength. Nonetheless, surface treatment improves the dispersion of SiO_2_ within the matrix. Notably, the composite containing 1.0 wt% of surface hydrophobic SiO_2_ exhibits the least space charge accumulation. Compared to the base material PP, the average charge density is reduced by 83.9% after the 1800 s short-circuit discharges. Moreover, its DC breakdown field strength reaches 3.45 × 10^8^ V/m, surpassing pure PP by 19.4% and untreated SiO_2_/PP composites of the same proportion by 24.0%.

## 1. Introduction

DC long-distance transmission has always been a crucial element of the Power Internet of Things, offering distinct advantages over AC cables. These advantages include lower transmission energy losses, reduced voltage drops, and an enhanced resistance to electromagnetic interference. In particular, high voltage direct current (HVDC) transmission holds immense potential for large-scale power transmission systems such as submarine cables, DC motor-driven systems, and renewable energy systems. Within the realm of HVDC transmission modes, the utilization of direct current cables has gained traction owing to their favorable electrical and thermal properties [1,2,3,4,5]. The current insulation material widely used for cables is cross-linked polyethylene (XLPE). Although XLPE possesses low electrical properties such as dielectric loss, chemical resistance, and aging resistance, it faces challenges when applied to long-haul, high-voltage DC cable insulation. During the extrusion of XLPE insulation production, scorch product blockage can introduce a substantial number of impurities that significantly degrade the insulation performance of DC cables. Additionally, the manufacturing process of XLPE involves cross-linking, which consumes a significant amount of energy, incurs high costs, and poses environmental pollution challenges during cable disposal after service retirement [6,7]. Given the growing emphasis on environmental protection in recent years, there is an urgent need for a new material to replace XLPE [6,7]. Polypropylene (PP) is a promising candidate as it possesses a melting point above 150 °C and can withstand long-term operating temperatures of 90 °C [8]. PP has a higher melting point than XLPE, making it suitable for cable insulation in high-temperature operations. It offers advantages such as high mechanical strength and does not require cross-linking during production, making it cost-effective and easier to recycle. As an environmentally friendly insulation material, PP is an excellent thermoplastic material known for its exceptional heat resistance [9,10,11]. However, its poor flexibility, inadequate insulating properties, and susceptibility to aging have impeded its widespread use [12,13,14,15].

In recent years, researchers have discovered that the incorporation of inorganic nanostructures into polymer matrices can effectively enhance the insulation performance of polymers [16,17,18,19,20,21]. Studies by Jain have explored the crystallization behavior of PP/SiO_2_ composites prepared through solid-state and in-situ methods, indicating that in-situ synthesized nanostructured SiO_2_ particles act as heterogeneous nucleating agents [22]. Fuse et al. found that certain nano-clay particles introduce ionic groups into the polypropylene matrix, which aggravates the accumulation of space charge in the composites. Hence, the dispersion treatment of nanostructures also significantly influences the insulation performance [23]. L. Cheng et al. investigated the breakdown mechanism in polypropylene-based nano dielectrics and revealed that the interface effect of nanostructures improves the energy level distribution of the semicrystalline polymer, reducing the probability of electrons reaching energies that exceed the breakdown activation energy and increasing the breakdown field strength [24]. Scholars Y. Suzuoki et al. identified space charge accumulation as one of the factors leading to reduced electrical strength in polypropylene. The gradual accumulation of space charge distorts the electric field applied to polypropylene, thereby reducing its electrical strength and potentially causing insulation breakdown in severe cases [25]. The accumulation of space charge in PP accelerates insulation aging and even breakdown, posing safety hazards in cable operations [26,27,28]. Thus, space charge accumulation poses a significant challenge and detrimentally affects the long-term reliability of PP cable operation.

Previous studies have confirmed that the addition of inorganic nanostructures can enhance the comprehensive properties of polymers to a certain extent. However, there is limited research on the DC space charge characteristics of SiO_2_-modified polypropylene. In this paper, SiO_2_/PP nanocomposites were studied by incorporating nanostructured SiO_2_ particles with different contents (0.5 wt%, 1.0 wt%, and 3.0 wt%) and subjecting them to different surface treatments (one without treatment and one with a hydrophobic surface treatment). The space charge characteristics and DC breakdown performance of the SiO_2_/PP nanocomposites were investigated, with two types of SiO_2_ particles serving as controls. The influence of nanoparticle surface treatment on the properties of composite materials is discussed, offering valuable insights for the application of polypropylene nanocomposites in cable insulation.

## 2. Materials and Methods

### 2.1. Materials

PP (4874, Borealis AG, Vienna, Austria)—density 912 kg/m^3^; melt flow rate—2.8 g/10 min; Nordic chemical nanostructured SiO_2_ (AEROSIL 200 and AEROSIL R974, Evonik Industries AG, Essen, Germany), of which AEROSIL 200 is the SiO_2_ without surface treatment (SiO_2_ 200) and AEROSIL R974 is hydrophobically treated with C_2_H_6_Cl_2_Si SiO_2_ (SiO_2_ R974).

The primary reaction process during the surface treatment of DDS is schematically shown in Figure 1. The hydrophilic hydroxyl groups in silica react with Cl in DDS to generate hydrophobic siloxane groups, completing the surface hydrophobicity treatment of SiO_2_.

SiO_2_/PP composites were prepared by the melting–blending method. The PP was dried at 60 °C for 24 h, and the SiO_2_ was dried at 80 °C for eight hours. The dried PP and two kinds of SiO_2_ were added to the torque rheometer according to the ratio in Table 1. The mixing temperature was 190 °C, the screw speed was 40 r/min, the mixing time was 20 min, and SiO_2_/PP nanocomposites were obtained. The prepared PP and SiO_2_/PP composite materials were filled into the mold and put into the plate vulcanizing machine. The temperature of the flat vulcanizer was set to 190 °C. The temperature of the flat vulcanizer was kept at 0 MPa for five minutes to make the sample melt completely. Then, the pressure was increased to 15 MPa (every 5 MPa was one stage), and each stage was kept for 5 min to obtain the required film sample.

### 2.2. FTIR Characterization

Fourier Transform infrared spectrometer (FTIR, Bruker AG, Germany) was used to measure SiO_2_ 200 and SiO_2_ R974 in the range of a 4000–400 cm^−1^ wave number. The FTIR spectra were used to study the effects of SiO_2_ hydrophobic treatment.

### 2.3. SEM Characterization

To assess the dispersion of nanostructures within the PP matrix, the composite’s morphology was examined using an SU8020 scanning electron microscope (SEM), manufactured by Hitachi in Japan. Before observation, all samples underwent freezing in liquid nitrogen to induce brittle fracture, followed by gold spraying on the fractured surface. The selected scale for observation was 1 μm.

### 2.4. Space Charge Test

The space charge distribution in the composites was assessed using pulsed electroacoustics (PEA). The SiO_2_/PP composite samples were subjected to a field strength of 4 × 10^7^ V/m for 30 min to induce the space charge distribution, which was then measured. Following this, the samples were grounded for 30 min, and the decay of the space charge in the samples was measured after a short-circuit discharge. To ensure optimal contact between the SiO_2_/PP composite sample and the electrode, silicone oil was applied. The experimental setup of PEA, as depicted in Figure 2, involved several components. The pulsed power supply provided a pulse voltage of 1 kV, a pulse width of 8 ns, and a repetition frequency of 2 kHz. The high voltage power supply supplied a range of a −20 × 10^3^ V to 20 × 10^3^ V DC high voltage. The signal coupling and sensing module offered a pulse time delay of more than 3 μs, a space charge sensitivity of 0.6 μC/m^3^, and a spatial resolution of 18 μm. Additionally, a preamplifier with a 400 MHz frequency response was employed.

### 2.5. Volume Resistivity Test

The resistivity of the specimens was measured using a three-electrode system and a pion meter under the action of a DC electric field with an electrode area of 2826 mm^2^, and the volume resistivity *ρ*_v_ at 4 × 10^7^ V/m was measured and calculated for each composite material using Equation (1).
(1)$$\rho_{V}=\frac{U}{I_{V}}\times \frac{\pi {\left(D_{1}+g\right)}^{2}}{4h}$$
where *ρ*_v_ is the volume resistivity, unit Ω·m; *I_V_* is the bulk current, unit A; *U* is the added DC voltage with a value of 8 × 10^3^ V; *h* is the thickness of the specimen with a value of 0.2 mm; $D_{1}$ is the diameter of the protected electrode, where the diameter of the electrode used in this experiment is 50 mm; *g* is the distance between the protected electrode and the measuring electrode, where the gap between the left and right sides is 5 mm.

### 2.6. Breakdown Strength Test

The DC breakdown performance test used the KZT-5 DC voltage breakdown testing machine produced by Yingkou Special Transformer Equipment Co., Ltd., Yingkou, China. The voltage was increased at a constant speed of l kV/s until the materials broke down. Two electrode systems were used in the DC breakdown experiment. The diameters of the upper and lower electrodes were 25 mm and 50 mm. To prevent surface discharge during the test, the sample and the whole electrode system needed to be wholly immersed in transformer oil. The electric breakdown strength field is generally analyzed by Weibull statistics of 2-parameter fitting measured data [29]. The formula is as follows:(2)$$P(E)=1-\exp[-{\left(E/E_{b}\right)}^{\beta }]$$
where *P*(*E*) is the cumulative probability of failure, *E* is the experimentally measured breakdown field strength, and *β* is the shape factor representing the dispersion of the data, when *E* = *E*_b_, then *P*(*E*) = 1 − e^−1^ = 0.632. So, regardless of how *β* varies, *E*_b_ is the breakdown field strength parameter with a cumulative breakdown probability of 63.2%. The 63.2% breakdown probability for solid insulation is considered in engineering to be the closest to the actual breakdown probability.

## 3. Results

### 3.1. Structural Characterization of Functional Groups

The absorption bands at 1105 cm^−1^ and 470 cm^−1^ are the asymmetric stretching vibrational band and bending vibrational band of Si–O in SiO_2_, respectively, while the symmetric stretching vibrational band of Si–O is at 799 cm^−1^, as shown in Figure 3. The strong and wide –OH band of SiO_2_ 200 in the wave number band of 3200–3700 cm^−1^ is due to the easy water absorption of SiO_2_. This is also due to the presence of a certain amount of loosely bound and tightly bound water on the surface of SiO_2_ 200 without surface hydrophobic treatment, while the intensity and width of the band of SiO_2_ R974 treated with a DDS surface hydrophobic treatment is significantly lower than that of SiO_2_ 200, indicating a significant reduction in the hydroxyl content of the hydrophilic groups on the surface and a significant reduction in surface-bound water.

### 3.2. SEM Morphology

Figure 4 displays the SEM images of the PP and SiO_2_/PP composites. When the samples undergo brittle fracture treatment, they experience stress, resulting in the appearance of wavy, striped areas in the images. In addition, the nanostructures are marked in the figure, where the positions with large agglomeration are circled by square boxes, and the relatively uniform positions are circled by column boxes. From the SEM images, it can be observed that in composites 0.5 US, 1.0 US, 0.5 TS, and 1.0 TS, the nanostructures are evenly dispersed within the PP matrix, and the agglomeration phenomenon is relatively minor. This is primarily due to the lower nano content in these composites.

However, in composites 3.0 US and 3.0 TS, severe agglomeration is evident. In composite 3.0 US, the agglomeration size reaches approximately 400 nm, which can be attributed to the high content of the untreated nanostructured SiO_2_. The presence of numerous hydrophilic hydroxyl groups in the material promotes aggregation. In composite 3.0 TS, the agglomeration size is around 150 nm. This is because the nanostructured SiO_2_ added to composite 3.0 TS undergoes surface treatment, where the hydroxyl groups on the surface of SiO_2_ react with silane to form a siloxane group. While most of the hydroxyl groups are removed, a small number of residual hydroxyl groups still exist, leading to agglomeration when the nano content is high. However, compared to composite 3.0 US, the agglomeration phenomenon is improved in composite 3.0 TS.

### 3.3. Space Charge Characteristics

The space charge distribution curves of the PP and SiO_2_/PP composites were measured at room temperature under a 4 × 10^7^ V/m polarization electric field for a duration of 30 min. The results are depicted in Figure 5. From the figure, it is evident that PP accumulates a significant amount of heteropolar charges at both the cathode and anode. The peak charge near the anode measures −11.40 C·m^−3^, while the peak charge near the cathode measures 11.74 C·m^−3^. The entire sample exhibits a substantial amount of space charge, although the accumulation of space charge in SiO_2_/PP composites is noticeably lower than in PP. However, there are distinct differences in their distribution.

In the case of the SiO_2_ 200/PP system, the accumulation of space charge in PP nanocomposites mixed with 0.5 wt% SiO_2_ 200 is effectively inhibited. With an increase in SiO_2_ 200 content, the space charge inside the sample gradually increases. For instance, when the nm content is 1.0 wt%, the maximum charge density near the cathode after 1800 s of polarization is 6.53 C·m^−3^. However, at a nano content of 3.0 wt%, charge accumulation occurs at both the anode and cathode, resulting in maximum charge densities of 5.12 C·m^−3^ and 4.23 C·m^−3^, respectively. Conversely, the addition of 0.5 wt% and 1.0 wt% SiO_2_ R974 into the PP matrix minimizes the accumulation of space charge. Among them, 1.0 wt% SiO_2_ R974 exhibits the strongest inhibitory effect on space charge, with only a small amount of space charge accumulating in the samples even after 1800 s of polarization. However, when the nano content is 3.0 wt%, the inhibitory effect on space charge weakens, and the maximum charge density after 1800 s reaches 4.90 C·m^−3^. Compared to PP, the peak value of space charge is reduced by 58.2%.

In summary, the results depicted in the figure demonstrate that the accumulation of space charge in SiO_2_/PP composites is lower than that in PP. The addition of nanomaterials effectively inhibits the accumulation of space charge. In particular, surface-treated SiO_2_ exhibits a superior inhibitory effect on space charge and minimizes charge accumulation. Notably, the inhibitory effect is prominent at the nano contents of 0.5 wt% and 1.0 wt%.

In Figure 6, the space charge distribution at various time intervals (30 s, 300 s, 600 s, 900 s, 1200 s, 1500 s, and 1800 s) after a short-circuit discharge is depicted. From the figure, it is evident that there is a significant presence of space charges at the cathode and anode of the PP sample after a short circuit, with the largest amount of residual charge. However, the addition of nanostructured SiO_2_ noticeably reduces the residual charge in the sample. When the nanostructured SiO_2_ content is 0.5 wt% and 1.0 wt%, the charge distribution of the two types of nanostructured SiO_2_ is similar, with a small amount of residual charge observed. On the other hand, when the nanostructured SiO_2_ content is 3.0 wt%, the residual charge is the highest, and there is a certain amount of positive and negative polarity charge at the cathode and anode. As the discharge time increases, the residual charge of all nanocomposites gradually decays to zero. However, even after the 1800 s of discharge, PP still retains a certain amount of residual charge.

To further analyze the internal residual charge of the PP and SiO_2_/PP composites after 30 min short-circuit discharges, the average volume charge density was used to quantitatively analyze the amount of space charge accumulated in the different samples, and its calculation formula is [30]:(3)$$Q(t;E_{t})=\frac{1}{x_{2}-x_{1}}\int_{x_{1}}^{x_{2}}\left|Q(x,t;E_{t})\right|dx$$
where *X*_2_ and *X*_1_ are the positions of the cathode and anode, respectively; *t* is the short-circuit time after removing the polarizing electric field; and *Q* (*x*, *t*; *E_t_*) is the space charge density inside the sample at any time and any position.

After the short-circuit discharge at 1800 s, the average volume charge density of space charge inside the PP and SiO_2_/PP composites is shown in Figure 7. As can be seen from the figure, the average charge density of residual charge in the PP material is the largest. The average charge density reaches 0.953 C·m^−3^ even after an 1800 s discharge. After nanostructured SiO_2_ was added, the average charge density of all materials decreased. The average charge density of the composite with SiO_2_ R974 was lower than that of the composite with SiO_2_ 200 on the whole, and the average charge density was the lowest when the SiO_2_ R974 content was 1.0 wt%, which was only 0.153 C·m^−3^. Compared with PP, it decreased by 83.9%.

### 3.4. Volume Resistivity

The volume resistivity of each specimen at a DC field of 4 × 10^7^ V/m reveals that nanostructured SiO_2_ effectively enhances the volume resistivity of PP composites. In Figure 8, among the PP composites doped with SiO_2_ 200, specimen 0.5 US exhibits the highest volume resistivity of 1.86 × 10^14^ Ω·m. However, the volume resistivity of specimen 3.0 US is less effective in enhancing the volume resistivity of PP due to the decrease in the specific surface area of nanostructured SiO_2_ caused by agglomeration.

In contrast, the enhancement of the volume resistivity of PP by SiO_2_ R974 is more pronounced compared to SiO_2_ 200. In the PP composite system with SiO_2_ R974, the volume resistivity of the PP composites with 0.5 wt% and 1.0 wt% performs well in the matrix. Specifically, specimen 0.5 TS exhibits a resistivity of 2.25 × 10^14^ Ω·m, while specimen 1.0 TS shows a resistivity of 7.41 × 10^14^ Ω·m. This can be attributed to the more uniform dispersion of SiO_2_ R974 nanostructures in the matrix after surface hydrophobic treatment. The improved interfacial interaction between SiO_2_ and the PP matrix enhances the compatibility between the inorganic nanofillers and the polymer matrix. The interfacial domains formed by this interaction are also more uniformly distributed, which introduces a scattering effect on the free-moving electrons and limits the rapid movement of carriers in the matrix. In Yao Zhou’s study on the effect of the space charge accumulation of titanium oxide nanostructures in PP composites, it was found that the presence of titanium oxide increased the number of shallow traps. This increase in shallow traps inhibited the accumulation of space charges and, as a result, enhanced the mobility of charge carriers. Consequently, the conductivity of the composites increased, leading to a decrease in volume resistivity [31].

### 3.5. DC Breakdown Strength

The breakdown strength test results of the PP and SiO_2_/PP composites were fitted using the Weibull distribution, and the findings are presented in Figure 9. The breakdown field strength results fitted by Weibull are summarized in Table 2. Based on the results of the DC breakdown experiment, it is observed that when using SiO_2_ 200 as the nanofiller with a nano content of 0.5 wt%, the DC breakdown field strength is the highest, exhibiting a 15.30% increase compared to pure PP. However, when the SiO_2_ content is increased to 1.0 wt% and 3.0 wt%, the DC breakdown field strength decreases, with the greatest reduction of 25.25% observed at 3.0 wt%. Similarly, when SiO_2_ R974 is used as the nanofiller at 0.5 wt% and 1.0 wt%, the DC breakdown field strength increases by 18.18% and 19.40%, respectively. In experiments conducted by Zhe Li to investigate the space charge and electrical strength of MgO nanostructures/PP composites, it was found that while MgO nanostructures can effectively inhibit charge injection into PP composites, the DC breakdown field strength of PP composites decreases as the particle concentration increases [32]. The comparison shows that SiO_2_ has a positive effect on the DC breakdown field strength of PP composites.

However, at a nano content of 3.0 wt%, the DC breakdown field strength of both nanocomposites decreases. Nevertheless, the DC breakdown field strength of the SiO_2_ R974/PP composite still exhibits a 30.58% increase compared to the SiO_2_ 200/PP composite. This can be attributed to the agglomeration of nanostructures in the matrix as the nano content increases. SiO_2_ 200, without surface treatment, exhibits poor compatibility with the PP matrix, leading to a more severe agglomeration in the matrix. This agglomeration acts as large particle impurities, increasing the heterogeneity of the material and significantly reducing the breakdown field strength. On the other hand, the addition of SiO_2_ R974 helps improve the agglomeration phenomenon. As a result, even at high nano content, the DC breakdown field strength only experiences a slight decrease. In contrast, Timothy Krentz investigated the impact of anthracene surface-modified SiO_2_ on the breakdown characteristics of PP composites. The findings revealed that at a low level, the use of treated SiO_2_ enhanced the breakdown field strength of PP composites. However, at a high level, there was a reduction of 15% in the breakdown field strength [33].

## 4. Discussion

According to space charge theory [34], space charge in polymers can originate from three main sources: electrode injection, impurity ion dissociation, and dipole orientation. Among these, electrode injection is the primary source of space charge. Observing the space charge distribution in the PP and SiO_2_/PP composites, as well as the residual charge after short-circuiting discharge, it becomes apparent that PP exhibits significant space charge accumulation under high field strengths of 4 × 10^7^ V/m. This is due to the rapid migration of electrode-injected charges and impurity-dissociated charges towards the poles, resulting in a higher accumulation of heteropolar charges near the electrodes. These heteropolar charges strengthen the electric field near the electrodes, thereby intensifying space charge accumulation [35]. Notably, even after short-circuit discharge, a considerable amount of residual charge remains in the samples.

By doping SiO_2_ particles into the PP matrix, numerous interfacial regions are introduced, which restrict the movement of electrons and contribute to the good bulk resistivity exhibited by the SiO_2_/PP composite [36,37]. Additionally, the presence of SiO_2_ nanostructures causes the scattering of carriers, reducing their mobility and mitigating charge accumulation. During the pressurization process, the SiO_2_ particles near the electrode trap polar charges, forming a shielding layer that weakens the electric field in the vicinity of the electrode and reduces charge injection to some extent. Consequently, the space charge within the nanocomposite is reduced.

The lower space charge accumulation and average charge density after discharge in the composites incorporating SiO_2_ R974 compared to those incorporating SiO_2_ 200 can be primarily attributed to the surface treatment of SiO_2_ with DDS. This treatment renders the siloxane chains and methyl groups formed by the reaction hydrophobic in nature. Consequently, the nanostructures are brought closer to the non-polar PP structure, facilitating better binding of the inorganic particles to the polymer matrix. As a result of the improved binding, the dispersion of nanomaterials within the matrix is enhanced, leading to tighter interfaces with the PP molecular chains. This, in turn, reduces defects in the PP structure, such as micron-sized cavities, and mitigates the damage caused to the PP molecular chains by high-energy electrons. The good dispersion of nanostructures provides more charge-trapping sites within the composite, and this is confirmed by the increased volume resistivity [19]. Ultimately, these factors contribute to an improvement in the DC breakdown field strength of the composite [38]. The schematic diagram of the SiO_2_ limiting carrier migration mechanism is shown in Figure 10.

## 5. Conclusions

Based on the findings from the SEM images, the PEA analysis, the volume resistivity test, and the DC breakdown tests, the following conclusions can be drawn from this paper:SEM images reveal that when the nanoparticle content reaches 3.0 wt%, agglomeration occurs in the matrix, which introduces impurities and reduces the breakdown field strength. However, with the addition of SiO_2_ R974, the surface hydrophobic treatment greatly reduced the hydroxyl content and improved the agglomeration phenomenon, resulting in a 30.58% increase in breakdown field strength and an enhanced space charge suppression compared to SiO_2_ 200 at the same ratio.The addition of a small amount of nanostructured SiO_2_ effectively inhibits space charge accumulation, weakens electrode charge injection, significantly reduces residual charge after short-circuiting discharge, and enhances charge capture capability. The hydrophobic siloxane group formed after surface treatment improves the compatibility between SiO_2_ and non-polar PP and reduces structural defects in the amorphous zone. Consequently, SiO_2_ R974 exhibits lower charge accumulation and residual charge after discharge compared to SiO_2_ 200. The composite with 1.0 wt% SiO_2_ R974 shows the least space charge accumulation and residual charge after discharge, with an average charge density of only 0.153 C·m^−3^ after 1800 s of a short-circuit discharge, which is 83.9% lower than that of PP.The addition of a small amount of nanostructured SiO_2_ into the PP matrix increases the DC breakdown field strength. When 0.5 wt% SiO_2_ 200 is added, the breakdown field strength increases by 15.3% compared to pure PP. Similarly, the addition of SiO_2_ R974 at 0.5 wt% and 1.0 wt% leads to an increase of 18.2% and 19.4% in the breakdown field strength, respectively, compared to PP. The surface treatment reduces the hydrophilicity of nanomaterials, improves their bonding with the polymer matrix, forms a higher polarity interface, and enhances dispersion. Therefore, SiO_2_ R974 exhibits a more significant effect in improving the breakdown field strength of PP.

By doping the PP material with an appropriate amount of SiO_2_ in the form of a DDS surface treatment, the accumulation of space charge in the composite can be effectively suppressed, and the breakdown field strength can be improved. This provides an excellent solution for the preparation of green cables.

## Figures and Tables

**Figure 1 polymers-15-02826-f001:**
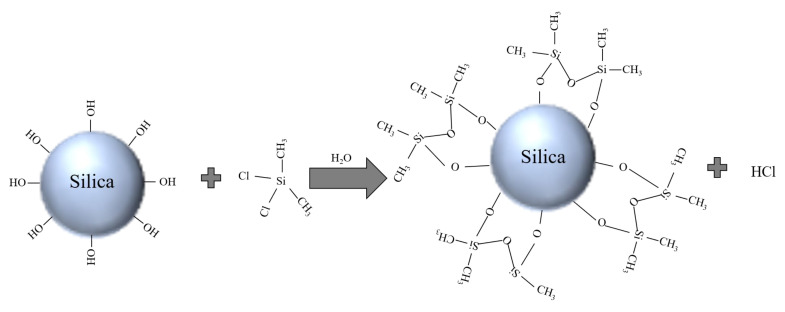
Schematic diagram of SiO_2_ surface treated with DDS.

**Figure 2 polymers-15-02826-f002:**
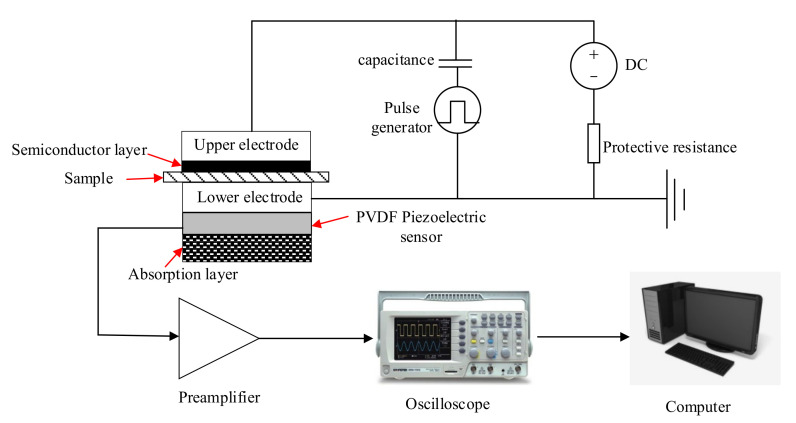
Schematic diagram of the experimental device of the space charge test system.

**Figure 3 polymers-15-02826-f003:**
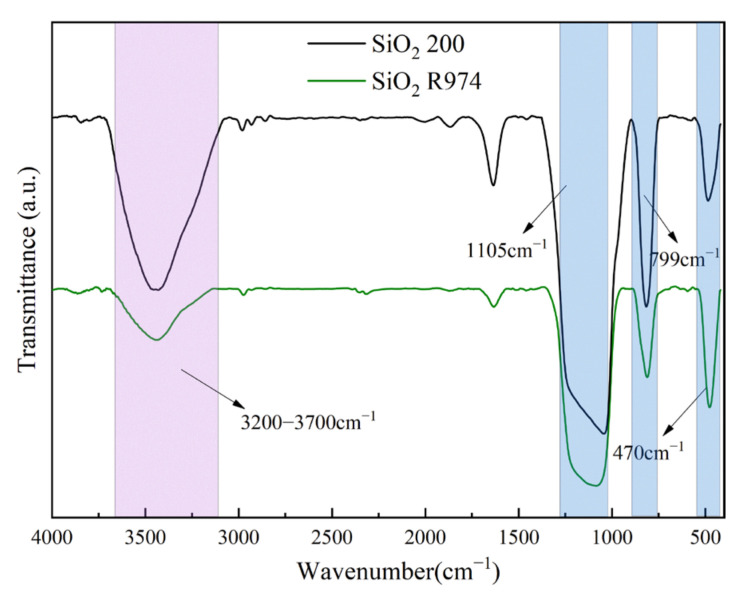
Fourier infrared spectra of SiO_2_ R974 and SiO_2_ 200.

**Figure 4 polymers-15-02826-f004:**
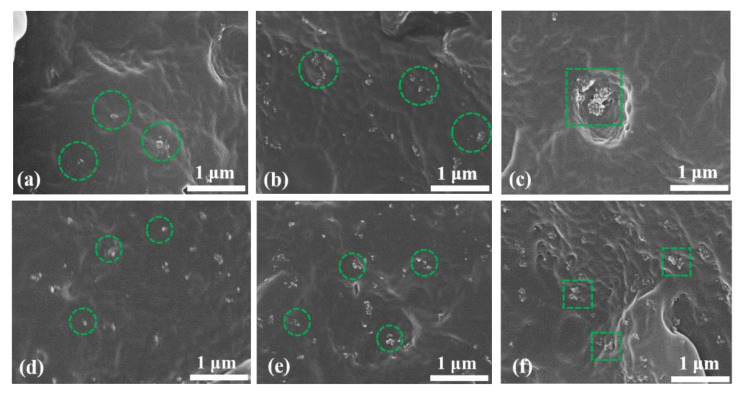
SEM images of PP and SiO_2_/PP composites: (**a**) 0.5 US, (**b**) 1.0 US, (**c**) 3.0 US, (**d**) 0.5 TS, (**e**) 1.0 TS, (**f**) 3.0 TS.

**Figure 5 polymers-15-02826-f005:**
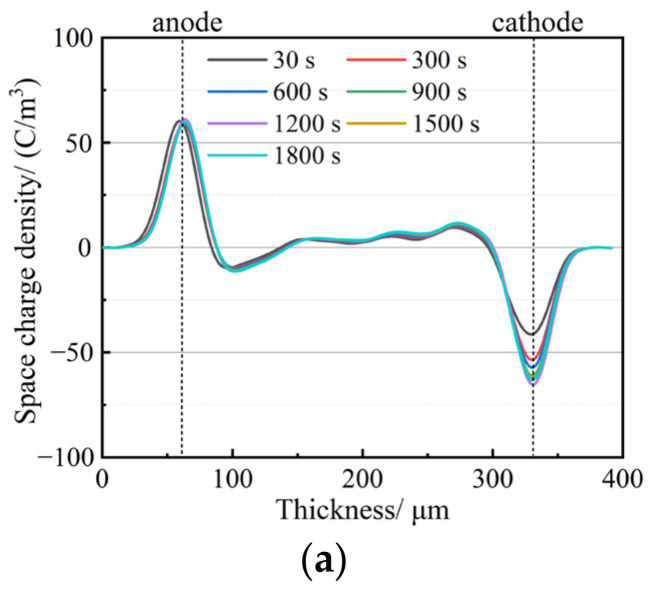
Space charge distribution of SiO_2_/PP with different kinds of composites under an electric field: (**a**) PP, (**b**) 0.5 US, (**c**) 1.0 US, (**d**) 3.0 US, (**e**) 0.5 TS, (**f**) 1.0 TS, (**g**) 3.0 TS.

**Figure 6 polymers-15-02826-f006:**
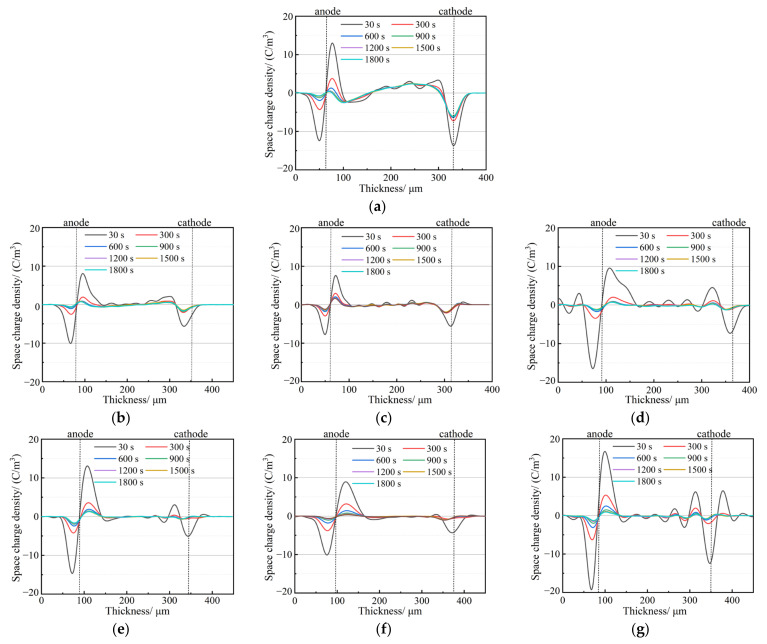
Space charge distributions of PP and SiO_2_/PP composites in short connection: (**a**) PP, (**b**) 0.5 US, (**c**) 1.0 US, (**d**) 3.0 US, (**e**) 0.5 TS, (**f**) 1.0 TS, (**g**) 3.0 TS.

**Figure 7 polymers-15-02826-f007:**
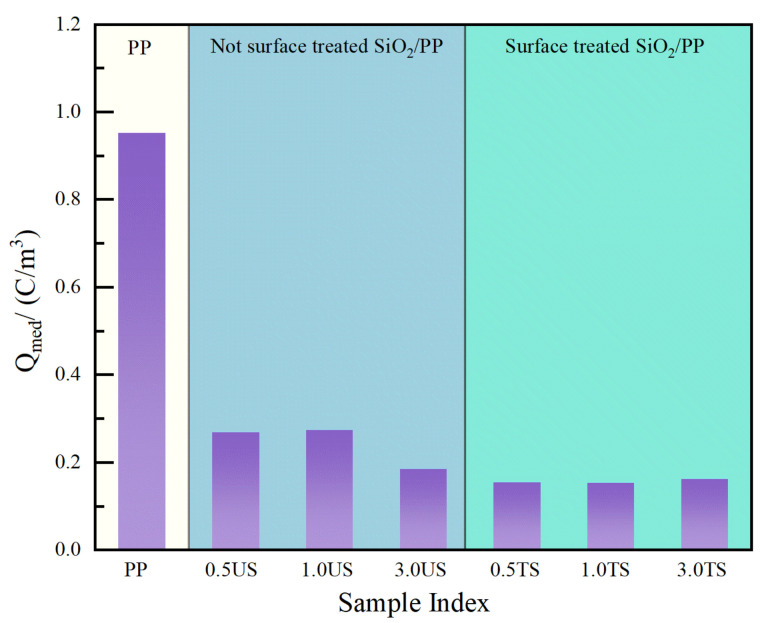
The average volume charge density of space charge in PP and SiO_2_/PP composites after 1800 s of a short-circuit discharge.

**Figure 8 polymers-15-02826-f008:**
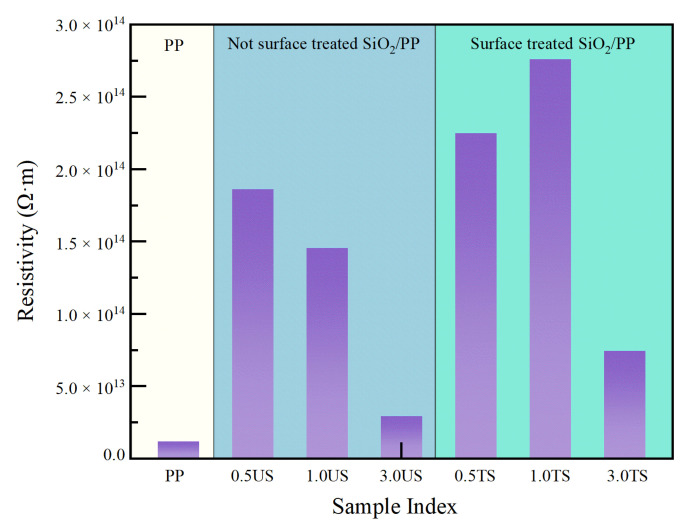
Volume resistivity of PP and SiO_2_/PP composites.

**Figure 9 polymers-15-02826-f009:**
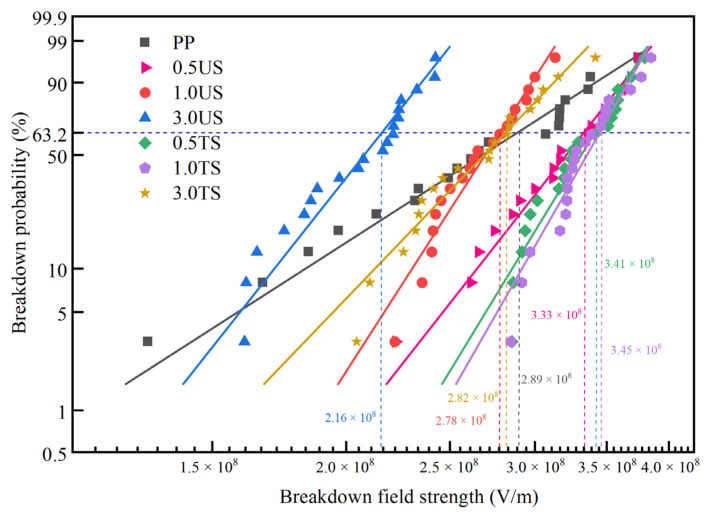
Fitting results of Weibull distribution parameters.

**Figure 10 polymers-15-02826-f010:**
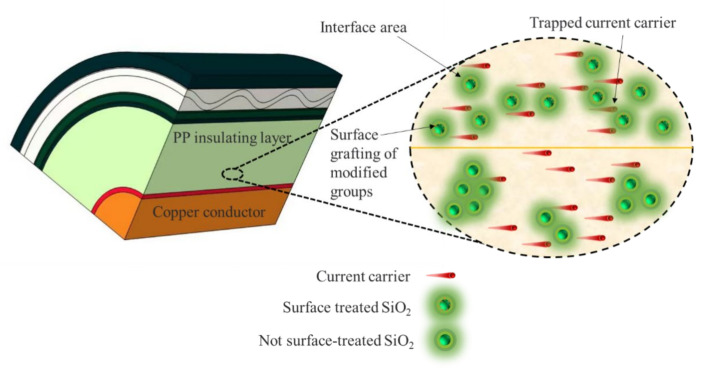
Schematic diagram of SiO_2_ limiting current carrier migration.

**Table 1 polymers-15-02826-t001:** The mass ratio (%) of SiO_2_/PP composites.

Specimen	PP/wt%	SiO_2_ 200/wt%	SiO_2_ R974/wt%
PP	100.0	0.0	0.0
0.5 US	99.5	0.5	0.0
1.0 US	99.0	1.0	0.0
3.0 US	97.0	3.0	0.0
0.5 TS	99.5	0.0	0.5
1.0 TS	99.0	0.0	1.0
3.0 TS	97.0	0.0	3.0

Note: “US” stands for untreated SiO_2_, and “TS” stands for treated SiO_2_.

**Table 2 polymers-15-02826-t002:** DC breakdown field strength of different composites.

Specimen	DC Breakdown Field Strength *E*_b_ (V/m)	Shape Parameter *β*
PP	2.89 × 10^8^	4.95
0.5 US	3.33 × 10^8^	9.83
1.0 US	2.78 × 10^8^	12.01
3.0 US	2.16 × 10^8^	9.76
0.5 TS	3.41 × 10^8^	12.69
1.0 TS	3.45 × 10^8^	13.50
3.0 TS	2.82 × 10^8^	8.02

## Data Availability

The data presented in this study are available on request from the corresponding author.

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
