# Peer review of "Space Charge Characteristics and Breakdown Properties of Nanostructured SiO2/PP Composites"

_polymers, 2023, doi:10.3390/polym15132826_

Round 1

Reviewer 1 Report

The proposal is interesting and of value to the scientific community, there are some considerations to be enriched, which are indicated below:

1.- It is recommended to add a list of acronyms.
2.- Indicate the parameters of the pulse generator, DC value and preamplifier used in the space charge test.
3.- Pg 4. Indicate the values of the variables: U, Iv and D1.
4.- Indicate why we worked with a failure probability value of 63.2%, I think it has to do with the probability density.
5.- Add a comparative table with respect to other techniques reported in the literature.

it is recommended that the document be read by an English speaker.

Author Response

Responses to Reviewers’ Comments

Dear Editors and Reviewers,

You have been a great help to our manuscript “Space Charge Characteristics and Breakdown Properties of Nano-SiO2/ PP Composites” (Manuscript Number: Polymers (ISSN 2073-4360). We have carefully revised this work based on these comments and suggestions. We will list every comment and answer them one by one. The comments were marked in blue, and the revisions were highlighted in red.

Thank you again for your time and consideration of our manuscript.

Sincerely,

Jun-Guo Gao

Reviewer #1

Questions and AnswersReviewer #1:

Comment 1: It is recommended to add a list of acronyms.

Response: Thank you for your suggestion. We have noted the material mass ratios for SiO2/PP material names, on page 3, line 115.

Table 1.  The mass ratio (%) of SiO2/PP composites.

specimen

PP/wt%

SiO2 200/wt%

SiO2 R974/wt%

PP

100.0

0.0

0.0

0.5US

99.5

0.5

0.0

1.0US

99.0

1.0

0.0

3.0US

97.0

3.0

0.0

0.5TS

99.5

0.0

0.5

1.0TS

99.0

0.0

1.0

3.0TS

97.0

0.0

3.0

Note: "US" stands for untreated SiO2, and "TS" stands for treated SiO2.

Comment 2: Indicate the parameters of the pulse generator, DC value and preamplifier used in the space charge test.

Response: A special thanks to you for your careful reading and professional advice on the details, on page 4, line 134. The space charge distribution in the composites was assessed using pulsed electroacoustics (PEA). The SiO2/PP composite samples were subjected to a field strength of 4 x 107 V/m for 30 minutes to induce the space charge distribution, which was then measured. Following this, the samples were grounded for 30 minutes, and the decay of the space charge in the samples was measured after a short circuit discharge. To ensure optimal contact between the SiO2/PP composite sample and the electrode, silicone oil was applied. The experimental setup of PEA, as depicted in Figure 2, involved several components. The pulsed power supply provided a pulse voltage of 1 kV, pulse width of 8 ns, and repetition frequency of 2 kHz. The high voltage power supply supplied a range of -20x103V to +20x103V DC high voltage. The signal coupling and sensing module offered a pulse time delay of more than 3 μs, space charge sensitivity of 0.6 μC/m3, and spatial resolution of 18 μm. Additionally, a preamplifier with a 400 MHz frequency response was employed.

Figure 2. Schematic diagram of the experimental device of space charge test system.

Comment 3: Pg 4. Indicate the values of the variables: U, Iv and D1.

Response: Thank you for your reminder. We have made the following changes and additions:

Where ρv is the volume resistivity, unit Ω·m; IV is the bulk current, unit A; U is the added DC voltage of 8×103V; h is the thickness of the specimen of 0.2mm;  is the diameter of the protected electrode, the diameter of the electrode used in this experiment is 50 mm; g is the distance between the protected electrode and the measuring electrode, the gap between the left and right sides is 5 mm.

Comment 4: Indicate why we worked with a failure probability value of 63.2%, I think it has to do with the probability density.

Response: Thanks for your meaningful comments. We have added the following clarification on the issue of breakdown probability.

Where P(E) is the cumulative probability of failure, E is the experimentally measured breakdown field strength and β is the shape factor representing the dispersion of the data, when E=Eb, then P(E) =1-e-1=0.632. So regardless of how β varies, Eb is the breakdown field strength parameter at a cumulative breakdown probability of 63.2%. The 63.2% breakdown probability for solid insulation is considered in engineering to be the closest to the actual breakdown probability.

Comment 5: Add a comparative table with respect to other techniques reported in the literature.

Response: We offer our thanks and admiration to you for your professional comment, which not only ensured the correct description of the theory but also promoted the improvement of the study and made the experimental results more reliable. We have compared this with other literature techniques on the suppression of space charge effects by PP-doped nanoparticles in the descriptions of bulk resistivity (page 10, lines 298-302), breakdown field strength (page 10, lines 313-316), and the effect of surface modification species (page 11, lines 331-334), and collated them into the following table. The following changes have been made to the manuscript in response to your comments, where the red font is the nanomaterial used in this paper:

Table. Comparison of the effects of doped nanoparticle types

and surface modification on PP properties

Types of doped particles

Effect of space charge

DC breakdown field strength effects

Carrier transport effects

TiO2

Moderate inhibitory effect

-

Increase

MgO

Better inhibition

Decrease

-

Anthracene surface modified with SiO2

-

Low levels increase,

high levels decrease

-

Surface treatment of SiO2 with DDS

Better inhibition

Low levels increase, almost constant at high levels

Decrease

References

  1. Zhou, Y.; Hu, J. Dang, B. Titanium oxide nanoparticle increases shallow traps to suppress space charge accumulation in polypropylene dielectrics. Rsc Advances, 2016, 6, 48720-48727.
  2. Li, Z.; Cao, W.; Sheng, G. Experimental study on space charge and electrical strength of MgO nano-particles/polypropylene composite. IEEE Transactions on Dielectrics and Electrical Insulation, 2016, 23, 1812-1819.
  3. Krentz, T.; Khani, M.M.; Bell, M. Morphologically dependent alternating‐current and direct‐current breakdown strength in silica–polypropylene nanocomposites. Journal of Applied Polymer Science, 2017, 134.

Reviewer 2 Report

Manuscript was well prepared, I have some minor corrections

- 2.1. Materials, include materials information, where purchased, etc. For example, SiO2 was purchased from ??, and purified with, used as received ….

- line 150, “vibration peak” change to “vibration band”. For FT-IR change “peak or peaks” to “band or bands”

- Figure 3 changes “Transmittance” to “Transmittance (a.u.)” Note. a.u. = arbitrary units

- Improve the explanation of FTIR results (lines 149-158)

- Figures have different format, try to prepare all Figures with the same format

Author Response

Responses to Reviewers’ Comments

Dear Editors and Reviewers,

You have been a great help to our manuscript “Space Charge Characteristics and Breakdown Properties of Nano-SiO2/ PP Composites” (Manuscript Number: Polymers (ISSN 2073-4360). We have carefully revised this work based on these comments and suggestions. We will list every comment and answer them one by one. The comments were marked in blue, and the revisions were highlighted in red.

Thank you again for your time and consideration of our manuscript.

Sincerely,

Jun-Guo Gao

Reviewer #2

Questions and AnswersReviewer #2:

Comment 1: 2.1. Materials, include materials information, where purchased, etc. For example, SiO2 was purchased from ??, and purified with, used as received ….

Response: Thank you for your reminder. The following changes have been made to the information on nanomaterials:

PP (4874, Borealis AG, Vienna, Austria) density 912 kg/m3, Melt flow rate 2.8 g/10 min, Nordic Chemical Nano SiO2 (AEROSIL 200, AEROSIL R974, Evonik Industries AG, Essen, Germany), of which AEROSIL 200 is SiO2 without surface treatment (SiO2 200), AEROSIL R974 is hydrophobically treated with C2H6Cl2Si SiO2 (SiO2 R974).

Comment 2: - line 150, “vibration peak” change to “vibration band”. For FT-IR change “peak or peaks” to “band or bands”

Response: We deeply appreciate your consideration and professional questions about the data. The following changes have been made to the information.

The absorption bands at 1105 cm-1 and 470 cm-1 are the asymmetric stretching vibrational band and bending vibrational band of Si-O in SiO2 respectively, while the symmetric stretching vibrational band of Si-O is at 799 cm-1, in picture 3. The strong and wide -OH band of SiO2 200 in the wave number band of 3200-3700 cm-1 is due to the easy water absorption of SiO2. This is due to the easy absorption of water by SiO2 and the presence of a certain amount of loosely bound and tightly bound water on the surface of SiO2 200 without surface hydrophobic treatment, while the intensity and width of the band of SiO2 R974 treated with DDS surface hydrophobic treatment is significantly lower than that of SiO2 200, indicating a significant reduction in the hydroxyl content of the hydrophilic groups on the surface and a significant reduction in surface-bound water.

Comment 3: - Figure 3 changes “Transmittance” to “Transmittance (a.u.)” Note. a.u. = arbitrary units

Response: Your suggestion is very helpful to us. And we have corrected the FTIR figure.

Figure 3. Fourier infrared spectra of SiO2 R974 and SiO2 200

Comment 4: - Improve the explanation of FTIR results (lines 149-158)

Response: Thank you for your suggestion. We have added to the data description section of the FTIR.

The absorption bands at 1105 cm-1 and 470 cm-1 are the asymmetric stretching vibrational band and bending vibrational band of Si-O in SiO2 respectively, while the symmetric stretching vibrational band of Si-O is at 799 cm-1, in picture 3. The strong and wide -OH band of SiO2 200 in the wave number band of 3200-3700 cm-1 is due to the easy water absorption of SiO2. This is due to the easy absorption of water by SiO2 and the presence of a certain amount of loosely bound and tightly bound water on the surface of SiO2 200 without surface hydrophobic treatment, while the intensity and width of the band of SiO2 R974 treated with DDS surface hydrophobic treatment is significantly lower than that of SiO2 200, indicating a significant reduction in the hydroxyl content of the hydrophilic groups on the surface and a significant reduction in surface-bound water.

Comment 5: -Figures have different format, try to prepare all Figures with the same format

Response: We offer our thanks to you once again for your thoughtful comments on the details and careful reading of this study. And we have formatted the figures in the paper to be uniform and aesthetically pleasing.

Reviewer 3 Report

The paper presents the study about space charge characteristics and breakdown properties of nano-SiO2/ PP composites. Authors present investigation of effect and mechanism of SiO2 with DDS surface hydrophobic treatment on space charge suppression and electrical properties of PP composites. They obtained results, which say, that incorporation of SiO2 nanoparticles effectively mitigates charge accumulation in PP composites. But high concentration of mentioned nanoparticles tends to space charge suppression and diminished DC breakdown field strength. On the other hands, surface treatment improves the dispersion of SiO2 within the matrix.

Dear author, thank you very much for interesting paper about PP, which is still important material, especially as insulation of high voltage cables, used in electric power systems. I put some comments and questions.

Comments:

1. Introduction chapter is well organized. Anyway, I would expect some information when DC lines are used in place of AC lines, if paper describes PP used in DC cables.

2. Authors could more detail describe the differences between PP and XLPE indicating advantages and disadvantages both materials. Please complete.

3. formula (1) – I think, it is much better to use SI unit system, then kV and mm. so, please change to [V] and [m].

4. Breakdown voltage probability results are very interesting. Anyway, the pure PP has average characteristics comparing to the studied cases in presented paper. It means that is some optimal concentration and type of nano particles in pure PP. it is very interesting conclusions, which may be used in commercial application, especially in electric power devices, see cable.

Author Response

Responses to Reviewers’ Comments

Dear Editors and Reviewers,

You have been a great help to our manuscript “Space Charge Characteristics and Breakdown Properties of Nano-SiO2/ PP Composites” (Manuscript Number: Polymers (ISSN 2073-4360). We have carefully revised this work based on these comments and suggestions. We will list every comment and answer them one by one. The comments were marked in blue, and the revisions were highlighted in red.

Thank you again for your time and consideration of our manuscript.

Sincerely,

Jun-Guo Gao

Reviewer #3

Questions and AnswersReviewer #3:

Comment 1: Introduction chapter is well organized. Anyway, I would expect some information when DC lines are used in place of AC lines, if paper describes PP used in DC cables.

Response: Thanks for your meaningful comments. At the same time, we sincerely apologize for the lack of information on the replacement of AC cables by DC cables. The following additions and changes have therefore been made to the beginning of the introduction.

DC long-distance transmission has always been a crucial element of the power Internet, offering distinct advantages over AC cables. These advantages include lower transmission energy losses, reduced voltage drops, and enhanced resistance to electromagnetic interference. In particular, high voltage direct current (HVDC) transmission holds immense potential for large-scale power transmission systems like submarine cables, DC motor-driven systems, and renewable energy systems. Within the realm of HVDC transmission modes, the utilization of direct current cables has gained traction owing to their favorable electrical and thermal properties [1-5].

Comment 2: Authors could more detail describe the differences between PP and XLPE indicating advantages and disadvantages both materials. Please complete.

Response: Your suggestion is very helpful to us. We, therefore, present a more detailed comparison of the advantages and disadvantages of the two materials. And three new references have been added. Particular emphasis is placed on the hidden problems of long-term operation after the application of PP to DC cables on page 2, lines 52 to 60.

The current insulation material widely used for cables is cross-linked polyethylene (XLPE). Although XLPE possesses low electrical properties such as dielectric loss, chemical resistance, and aging resistance, it faces challenges when applied to long-haul, high-voltage DC cable insulation. During the extrusion of XLPE insulation production, scorch product blockage can introduce a substantial amount of impurities that significantly degrade the insulation performance of DC cables. Additionally, the manufacturing process of XLPE involves cross-linking, which consumes a significant amount of energy, incurs high costs, and poses environmental pollution challenges during cable disposal after service retirement [6-7]. Given the growing emphasis on environmental protection in recent years, there is an urgent need for a new material to replace XLPE [6-7]. Polypropylene (PP) is a promising candidate as it possesses a melting point above 150°C and can withstand long-term operating temperatures of 90°C [9]. PP has a higher melting point than XLPE, making it suitable for cable insulation in high-temperature operations. It offers advantages such as high mechanical strength and does not require cross-linking during production, making it cost-effective and easier to recycle. As an environmentally friendly insulation material, PP is an excellent thermoplastic material known for its exceptional heat resistance [10-12]. However, its poor flexibility, inadequate insulating properties, and susceptibility to aging have impeded its widespread use [13-16].

The gradual accumulation of space charge distorts the electric field applied to polypropylene, thereby reducing its electrical strength and potentially causing insulation breakdown in severe cases [26]. The accumulation of space charge in PP accelerates insulation aging and even breakdown, posing safety hazards in cable operations [27-29]. Thus, space charge accumulation poses a significant challenge and detrimentally affects the long-term reliability of PP cable operation.

References

  1. Li, G.; Gu, Z.; Xing, Z. Space Charge and Trap Distributions and Charge Dynamic Migration Characteristics in Polypropylene under Strong Electric Field. ECS Journal of Solid State Science and Technology, 2022, 11, 083003.
  2. Gao, J.G.; Liu, H.S.; Lee, T.T. Effect of Hydrophilic/Hydrophobic Nanostructured TiO2 on Space Charge and Breakdown Properties of Polypropylene. Polymers, 2022, 14, 2762.
  3. Zha, J.W.; Wu, Y.H.; Wang, S.J. Improvement of space charge suppression of polypropylene for potential application in HVDC cables. IEEE Transactions on Dielectrics and Electrical Insulation, 2016, 23, 2337-2343.

Comment 3: formula (1) – I think, it is much better to use SI unit system, then kV and mm. so, please change to [V] and [m].

Response: The breakdown Weibull diagram has been revised and the relevant kV/mm has been changed to V/m in the text.

Figure 9. Fitting results of Weibull distribution parameters

Comment 4: Breakdown voltage probability results are very interesting. Anyway, the pure PP has average characteristics comparing to the studied cases in presented paper. It means that is some optimal concentration and type of nano particles in pure PP. it is very interesting conclusions, which may be used in commercial application, especially in electric power devices, see cable.

Sincerely thank you for your high opinion of this study, this group will do further in-depth research in this direction.
